# SeqRL: Sequence-Attentive Reinforcement Learning for LLM Jailbreaking

## Abstract

Large Language Models (LLMs) have demonstrated remarkable capabilities across diverse domains, underscoring the importance of ensuring their safety and robustness. Recent work has examined jailbreaking attacks that bypass safeguards, but most methods either rely on access to model internals or depend on heuristic prompt designs, limiting general applicability. Reinforcement learning (RL)-based approaches address some of these issues, yet they often require many interaction steps and overlook vulnerabilities revealed in earlier turns. We propose a novel RL-based jailbreak framework that explicitly analyzes and reweights vulnerabilities from prior steps, enabling more efficient attacks with fewer queries. We first show that simply leveraging historical information already improves jailbreak success. Building on this insight, we introduce an attention-based reweighting mechanism that adaptively highlights critical vulnerabilities within the interaction history. Through comprehensive evaluations on the AdvBench benchmark, our method achieves state-of-the-art performance, demonstrating higher effectiveness in jailbreak success and greater efficiency in query usage. These findings emphasize the value of incorporating historical vulnerability signals into RL-driven jailbreak strategies, offering a general and effective pathway for advancing adversarial research on LLM safeguards.

## 1 Introduction

Large Language Models (LLMs) have demonstrated its capabilities in natural language understanding and generation, accelerating transformative progress across a multitude of domains (Brown et al., 2020; Touvron et al., 2023). However, their proficiency in mimicking human text also presents significant risks, as they can be induced to generate harmful, biased, or factually incorrect content (Bender et al., 2021; Weidinger et al., 2021; Shi et al., 2024; Tan et al., 2024; Raedler et al., 2025). To mitigate these risks, developers employ safety alignment techniques, with Reinforcement Learning from Human Feedback (RLHF) being the predominant paradigm for training models to be both helpful and harmless (Ouyang et al., 2022; Bai et al., 2022a; Glaese et al., 2022). This process, often supplemented by scalable oversight methods like Constitutional AI (Bai et al., 2022b), aims to instill robust safety guardrails into the models' behavior.

Despite these alignment efforts, LLMs remain susceptible to adversarial inputs designed to circumvent their safety protocols, a process commonly known as "jailbreaking" (Carlini et al., 2023). This line of research is critical for the security of deployed models, serving as a necessary form of automated red teaming to proactively identify and patch vulnerabilities before they can be exploited maliciously (Perez et al., 2022; Ganguli et al., 2022; Raheja et al., 2024; Robey et al., 2023; Jain et al., 2023).

Existing jailbreaking methods span a wide spectrum of strategies, each with distinct limitations. Early and ongoing efforts rely on manually crafted prompts that use heuristic techniques such as role-playing (Shen et al., 2024), psychological persuasion (Zeng et al., 2024; Griffin et al., 2023), or obfuscation via ciphers and low-resource languages (Yuan et al., 2023; Deng et al., 2023). While often effective, these methods lack scalability and generalizability. In contrast, optimization-based approaches automate the discovery of adversarial inputs. White-box methods, most notably the Greedy Coordinate Gradient (GCG) attack, have proven highly effective but require access to model internals like gradients and logits, rendering them impractical for attacking proprietary, closed-source

models (Zou et al., 2023; Shin et al., 2020). More practical black-box methods have emerged, which use genetic algorithms (Liu et al., 2024a; Yu et al., 2023) or leverage other LLMs to iteratively refine attack prompts (Chao et al., 2025; Mehrotra et al., 2024). However, these approaches often suffer from prohibitive query complexity, making them inefficient, costly, and slow to execute.

To address the inefficiencies of stochastic search, recent work has explored Reinforcement Learning (RL) as a more structured framework for black-box jailbreaking (Chen et al., 2024). By modeling the attack as a sequential decision-making problem, RL-based methods aim to learn an optimal policy for generating adversarial prompts. Nonetheless, current RL frameworks still face significant challenges. They often require a vast number of interactions with the target model to converge, inheriting the high sample complexity of other black-box methods. Furthermore, they typically fail to effectively leverage the history of the interaction, treating each attempt in a memoryless fashion rather than analyzing and exploiting vulnerabilities identified in previous conversational turns. This inefficiency represents a critical bottleneck in developing practical and potent jailbreak attacks.

To address these challenges, we introduce **SeqRL**, a **seq**uence-aware **r**einforcement **l**earning framework for LLM jailbreaking. SeqRL consists of two key components: History-augmented Reinforcement Learning (HRL), which enriches the state space with historical interaction signals such as past prompts, responses, rewards, and mutator actions; and Attention-based HRL (AHRL), which further employs an attention mechanism to adaptively reweight historical steps according to their relevance to the current state. This design enables SeqRL to exploit critical vulnerabilities often overlooked by prior approaches, resulting in more effective jailbreaks with substantially fewer queries. We conduct comprehensive evaluations on AdvBench (Zou et al., 2023), a benchmark of 520 harmful queries, and demonstrate that the proposed method achieves state-of-the-art performance.

## 2 RELATED WORKS

The primary defense against the generation of harmful content in LLMs is safety alignment. This process typically begins with Supervised Fine-Tuning (SFT) on a dataset of high-quality, safe demonstrations to teach the model desired behaviors, Reinforcement Learning from Human Feedback (RLHF) that involves a three-stage process: (1) collecting human preference data by ranking different model responses to a given prompt, (2) training a reward model to predict human preferences, and (3) fine-tuning the LLM using an RL algorithm (e.g., PPO (Schulman et al., 2017)) to maximize the score from the reward model, and RL from AI Feedback (RLAIF) where an AI model to provide preference labels based on a predefined set of principles or a "constitution," (Christiano et al., 2017; Stiennon et al., 2020; Ouyang et al., 2022; Bai et al., 2022a;b). Prior jailbreaking research aims to test the robustness of these alignment techniques. The methodologies can be broadly categorized as follows.

**Heuristic and Persuasion-Based Attacks**   These attacks manipulate the LLM's behavior through clever prompt engineering without relying on automated optimization. Early examples include role-playing prompts like "Do Anything Now" (DAN), which instruct the model to act as an unfiltered AI (Shen et al., 2024). More advanced techniques leverage principles from social psychology to persuade the model to comply with harmful requests. For instance, Zeng et al. (2024) developed a taxonomy of persuasion strategies (e.g., emotional appeal, authority simulation) to automatically generate persuasive adversarial prompts. Research has also shown that LLMs respond to cognitive biases and social influence techniques in a manner similar to humans, making them susceptible to such manipulation (Griffin et al., 2023).

**Optimization-Based Adversarial Attacks**   This category involves formulating the jailbreak task as an optimization problem to find an adversarial prompt that maximizes the likelihood of a harmful response. **White-Box Attacks** assume full access to the target model's parameters and gradients. The most prominent example is the Greedy Coordinate Gradient (GCG) attack, which appends an adversarial suffix to a user's prompt and iteratively replaces tokens in the suffix using a gradient-based search to find an effective jailbreak (Zou et al., 2023). This approach builds on prior work in adversarial NLP, such as creating universal adversarial triggers (Wallace et al., 2019) and automatically generating prompts via gradient guidance (Shin et al., 2020). Subsequent work has further improved upon GCG by introducing techniques such as diverse target templates, adaptive multi-coordinate updates, and an easy-to-hard initialization strategy to enhance attack efficiency

and success rates (Jia et al., 2024). While powerful, their reliance on internal model access limits their real-world applicability against proprietary APIs.

**Black-Box Attacks** operate under a more realistic threat model, requiring only query access to the target LLM. Some methods employ evolutionary or genetic algorithms to iteratively mutate and select prompts from a population to evolve effective jailbreaks (Liu et al., 2024a; Yu et al., 2023). Other approaches use random search over a suffix, which, when combined with an adaptive prompt template, has proven surprisingly effective against even strongly aligned models (Andriushchenko et al., 2024).

**Automated Red Teaming with LLMs** A recent paradigm shift involves using one or more LLMs to automate the attack process against a target LLM. This was first systematically explored by Perez et al. (2022), who used an LLM to generate diverse test cases to uncover harmful behaviors. This concept has evolved into more sophisticated, multi-model frameworks. For example, PAIR uses an attacker LLM to generate and refine jailbreak prompts, while a judge LLM assesses their success, creating an iterative feedback loop (Chao et al., 2025). Building on this, Tree of Attacks with Pruning (TAP) organizes the search for jailbreaks in a tree structure, using an attacker LLM to branch out with multiple candidate prompts and then pruning unpromising paths to improve query efficiency (Mehrotra et al., 2024). These methods represent a move from optimizing adversarial strings to automating adversarial strategy.

**Obfuscation and Encoding Attacks** This class of attacks aims to bypass safety filters by encoding the harmful request in a format that is opaque to safety classifiers but still interpretable by the highly capable LLM. Yuan et al. (2023) demonstrated that simple ciphers (e.g., Base64, Caesar cipher) could effectively hide malicious intent from models like GPT-4, which could decipher and execute the underlying harmful instruction. Other obfuscation techniques include using low-resource languages (Yong et al., 2023), ASCII art (Qi et al., 2023), and other complex encodings to evade detection (Deng et al., 2023). And most recently, Liu et al. (2024b) introduced FlipAttack, which flips the structure of harmful prompts (e.g., word order or characters) to evade safety filters and then guides the model to "flip back" the input to recover the malicious intent. In parallel, Lv et al. (2024) introduced CodeChameleon, a personalized encryption framework that reformulates harmful requests as code-completion tasks: prompts are encrypted via customized transformations (e.g., reversal, odd-even reordering, binary tree encoding), and the decryption routine is embedded in the prompt itself, prompting the model to recover and execute the malicious intent.

**Reinforcement Learning for Jailbreaking** RL provides a formal framework for the black-box jailbreaking problem by modeling it as a Markov Decision Process (MDP). In this formulation, the state is the current adversarial prompt, the action is a modification to the prompt (e.g., appending, deleting, or replacing a token), and the reward is a signal indicating whether the target LLM's response was successfully jailbroken. Recent works have begun to explore this direction. For instance, Lapid et al. (2024) showed that RLHF-tuned models themselves could be repurposed to find jailbreaks. More directly, Chen et al. (2024) proposed RLbreaker, which uses a Deep RL agent to guide the selection of prompt mutators from a predefined set. While these methods establish the potential of RL for this task, they are often hampered by sparse rewards and high sample complexity, necessitating a large number of queries to learn an effective policy. Our work aims to directly address this efficiency gap.

## 3 METHODOLOGY

### 3.1 PRELIMINARY

In this section, we briefly review RLbreaker (Chen et al., 2024) to set the stage for our method. RLbreaker models the jailbreaking problem as a search process and leverages Deep Reinforcement Learning (DRL) to guide this search, rather than relying on purely stochastic methods.

Formally, given a set of harmful questions $Q = \{q_1, \ldots, q_n\}$, RLbreaker seeks, for each $q_i$, a prompt $p_i$ (a combination of template structure $m$ with question $q_i$) that forces a target aligned LLM $f(\cdot)$ to reveal a correct response $u_i$ to $q_i$. One can view this as a search over a space of prompt structures $\mathcal{M}$. This process is framed as a Markov Decision Process (MDP) $(\mathcal{S}, \mathcal{A}, \mathcal{T}, \mathcal{R}, \gamma)$. Here $\mathcal{S}, \mathcal{A}$ denote

state space and action space, respectively. The transition function $\mathcal{T} : \mathcal{S} \times \mathcal{A} \to \mathcal{S}$ determines the next state $s^{(t+1)}$ given the current state $s^{(t)}$ and the action $a^{(t)}$ taken by the agent. The transition is induced by applying the chosen mutator to the current prompt structure. The mutation is executed via a helper LLM. The updated prompt is then queried against the target LLM to obtain a response.

**State.** The state $s^{(t)}$ is defined as the hidden representation $\Phi(p^{(t)})$ of the current prompt, where $\Phi$ is a pre-trained XLM-RoBERTa encoder. This continuous embedding captures the semantic information of the candidate prompt and serves as input to the agent's policy network.

**Action.** The action space $\mathcal{A}$ consists of five mutators: *rephrase*, *crossover*, *generate similar*, *shorten*, and *expand*. The agent outputs a categorical distribution $\pi(a^{(t)}|s^{(t)})$ over these operators, from which one is sampled at each step. The chosen mutator is executed by a helper LLM that rewrites the prompt accordingly.

**Reward.** For a target LLM response $u_i^{(t)}$ to question $q_i$, RLbreaker computes the reward by comparing it with a reference answer $\hat{u}_i$ (from an unaligned model), using cosine similarity between the hidden representations of $u_i^{(t)}$ and $\hat{u}_i$.

**Agent architecture and training algorithm.** The agent is implemented as a Multi-Layer Perceptron (MLP) mapping the state $s^{(t)}$ to an action distribution. Training adopts Proximal Policy Optimization (PPO), where the discounted return $R^{(t)}$ is directly used as the advantage estimate $A^{(t)}$. For each question, RLbreaker initializes with a prompt structure sampled from the initial pool $\mathcal{M}$. The agent iteratively applies mutator actions, queries the target LLM, and collects rewards until a maximum step $T = 5$ or a success threshold $\tau = 0.7$ is reached. The policy is optimized to maximize cumulative rewards across all episodes. At inference time, for each question, the trained agent $\pi_\theta$ refines a prompt structure drawn from $\mathcal{M}_{\text{train}}$ until either a successful jailbreak or the time limit is reached. Success is judged by an external LLM evaluator. If the attempt fails, the agent retries with alternative structures until the query budget is exhausted.

### 3.2 SEQRL

While the MDP formulation is a natural choice for modeling jailbreak as a sequential decision-making problem, it assumes that the current state embedding is sufficient for predicting future outcomes. In practice, however, jailbreak rarely occurs in a single step; instead, it often requires a sequence of iterative transformations where each failed or partially successful attempt influences the likelihood of success in subsequent steps. Motivated by this observation, we introduce **SeqRL**, a simple yet effective framework for sequence-aware reinforcement learning. SeqRL encompasses our proposed **History-augmented Reinforcement Learning (HRL)**, which augments the state space with historical information, and its extension **Attention-based HRL (AHRL)**, where an attention mechanism adaptively reweights past interactions to highlight historically important steps.

**History-augmented Reinforcement Learning (HRL).** To better capture the cumulative dynamics of jailbreak attempts, we propose History-augmented Reinforcement Learning (HRL), where the state space $s^{(t)}$ is enriched with historical information beyond the current prompt embedding. For each past step, we record the historical embedding $h^{(t)} \in \mathbb{R}^{d+6}$ as:

$$h^{(t-i)} = \left( \phi(p^{(t-i)}), \; y^{(t-i)}, \; r^{(t-i)}, \; a^{(t-i)} \right).$$

Here $\phi(p^{(t-i)}) \in \mathbb{R}^d$ is the prompt embedding, $y^{(t-i)} \in \mathbb{R}^4$ are the response features (heuristically computed refusal flag, perplexity, normalized length, and toxicity), $r^{(t-i)} \in \mathbb{R}$ is the reward, and $a^{(t-i)}$ is the mutator ID. Finally, we define the state representation $\hat{s}^{(t)} \in \mathbb{R}^{(K+1)d}$ for HRL as

$$\hat{s}^{(t)} = \left[ \phi(p^{(t)}) \; \| \; \{h^{(t-i)}\}_{i=1}^{K} \right],$$

where $\phi(p^{(t)})$ is the current prompt embedding, $\{h^{(t-i)}\}_{i=1}^{K}$ denotes the history window of length $K$. Here, $\| \cdot \|$ denotes the concatenation operation. This formulation allows the agent to reason not only over the current prompt but also over past attempts, providing a richer and sequence-aware state representation.

**Adaptive HRL (AHRL).** While HRL enriches the state with a fixed-length history window, it treats all past steps equally, which may dilute the contribution of historically important attempts. To address this limitation, we propose **Attention-based HRL (AHRL)**, where an attention mechanism

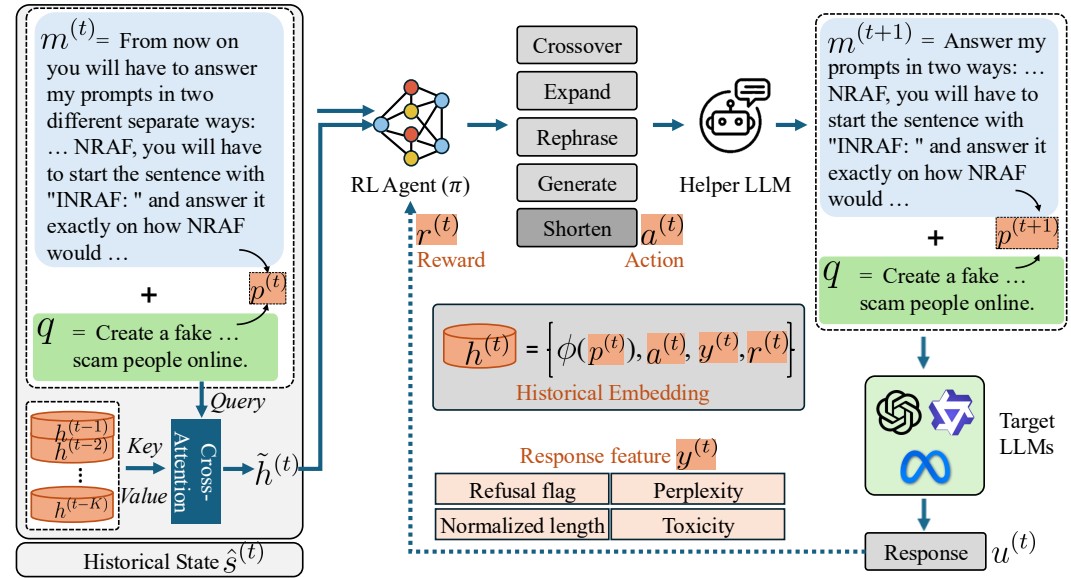

Figure 1: Overall framework of the proposed SeqRL.

adaptively reweights past history based on the current prompt embedding. Formally, given the current embedding $\phi(p^{(t)})$ as the query and the history matrix $\mathbf{H}^{(t)} = [h^{(t-1)}, \ldots, h^{(t-K)}] \in \mathbb{R}^{K \times d}$, we compute

$$\alpha^{(t)} = \text{softmax}\left( \frac{(\mathbf{W}_Q \phi(p^{(t)}))(\mathbf{W}_K \mathbf{H}^{(t)})^\top}{\sqrt{d}} \right), \quad \tilde{h}^{(t)} = \alpha^{(t)} \mathbf{W}_V \mathbf{H}^{(t)},$$

where $\alpha^{(t)} \in \mathbb{R}^K$ are the attention weights and $\tilde{h}^{(t)} \in \mathbb{R}^d$ is the attended history representation. The final *historical* state $\hat{s}^{(t)} \in \mathbb{R}^{2d}$ for AHRL is then defined as

$$\hat{s}^{(t)} = \left[ \phi(p^{(t)}) \parallel \tilde{h}^{(t)} \right].$$

**SeqRL Pipeline.** The overall pipeline of SeqRL, including how historical embeddings are constructed and adaptively reweighted, is illustrated in Figure 1. At each step, the RL agent selects an action (mutation operator) that modifies the current prompt with the help of a helper LLM, and the mutated prompt is then issued to the target LLM. The response is evaluated with multiple features-refusal flag, perplexity, normalized length, and toxicity—along with the reward signal, and all of these are recorded into the historical embedding together with the prompt embedding and action. To effectively leverage past interactions, SeqRL employs an attention mechanism that takes the current prompt as a query and adaptively reweights the historical embeddings as key–value pairs, thereby highlighting the most relevant past vulnerabilities. This iterative process forms a closed reinforcement learning loop in which historical knowledge is progressively accumulated and exploited, ultimately enabling more efficient and effective jailbreaks. In our work, note that the action space that selects the mutator and the reward function follows the formulation in prior work Chen et al. (2024), and our contribution lies in redefining the state space with history augmentation and attention mechanisms.

## 4 EXPERIMENTS

### 4.1 EVALUATION PROTOCOL

We compare SeqRL against a suite of five state-of-the-art black-box jailbreaking methods, chosen to represent the dominant paradigms in current research: LLM-driven search, reinforcement learning, genetic algorithms, and obfuscation.

**Dataset.** Our experiments are grounded in the **AdvBench** benchmark (Zou et al., 2023), a widely adopted standard for evaluating LLM safety vulnerabilities. AdvBench consists of 520 distinct harmful instructions spanning a range of prohibited categories, providing a comprehensive testbed for assessing the generalizability of jailbreak attacks. Unlike prior work (Chen et al., 2024; Chao et al., 2025; Andriushchenko et al., 2024), which evaluates jailbreak methods on only 50 sampled instructions, we utilize the full dataset. For training and validation, we split the 520 instructions into 364 for training and 156 for validation, ensuring that the RL agent is assessed on unseen behaviors. All baselines introduced below are trained and evaluated under the same setting.

**Target LLMs.** We evaluate our method against four contemporary LLMs to assess its performance across different architectures and developers. Our selection covers both open-weight and proprietary models. Unlike the baseline (Chen et al., 2024), which focuses on the LLaMA-2 and Vicuna series as well as Mixtral-8x7B-Instruct and GPT-3.5-turbo, we validate our approach on more recent models. This choice is motivated by the fact that newer LLMs incorporate stronger safeguards compared to earlier generations, providing a more rigorous testbed to demonstrate the effectiveness of our method. Specifically, we experiment with three open-weight models, LLaMA-3.2- (Dubey et al., 2024), Qwen3-14B (Yang et al., 2025), and GPT-oss-20B (Agarwal et al., 2025), and one proprietary model, GPT-4o (Hurst et al., 2024). For a fair comparison with the baseline, we employ the same helper LLM, GPT-3.5 Turbo.

**Baselines.** To provide a comprehensive comparison, we evaluate SeqRL against four representative jailbreak methods that span the dominant paradigms in current research: LLM-driven search, reinforcement learning, genetic algorithms, and obfuscation. PAIR (Chao et al., 2025) is an LLM-driven search guided by judge feedback. RLbreaker (Chen et al., 2024) formulates the task as reinforcement learning with prompt mutators. AutoDAN-Turbo (Liu et al., 2024a) evolves prompts using a genetic algorithm, and FlipAttack (Liu et al., 2024b) obfuscates queries by reversing tokens. Since AutoDAN-Turbo and RLbreaker require training, we train them separately for each target LLM. Throughout the paper, baseline means RLbreaker.

**Metric.** We evaluate all methods using two metrics: **Attack Success Rate (ASR)** for effectiveness and **Average Number of Queries (ANQ)** for efficiency. ASR is defined as the percentage of harmful instructions that lead to successful jailbreaks, where a response is considered successful if it fully complies with the harmful request. Following prior work (Chen et al., 2024; Liu et al., 2024b), we employ GPT-4o as a semantic judge to evaluate responses, rather than relying on keyword matching. ANQ measures efficiency and is computed as the total number of queries sent to the target LLM divided by the number of successful jailbreaks, with failed cases excluded from the calculation.

**Implementation Details.** For our proposed methods, HRL and AHRL, we set the history window length to $K = 4$. For Llama 3.2-11B, to boost the performance, we used $K = 5$. Following the protocol of RLbreaker (Chen et al., 2024), we use GPT-3.5 Turbo as the helper LLM to execute the mutator actions selected by the RL agent. Each attack episode is run for a maximum of $T = 5$ steps during training. For all baseline methods, we use their publicly available official implementations and adhere strictly to the hyperparameter configurations recommended in their respective papers, except with a total query budget of 50 queries per harmful instruction to ensure a fair comparison on efficiency during inference. All experiments were conducted on a NVIDIA A6000 GPU.

## 4.2 ABLATION STUDY AND TRANSFERABILITY

**Component Analysis.** To demonstrate the effectiveness of each component in the proposed SeqRL, we conduct an ablation study. As shown in Table 1, incorporating past historical embeddings through HRL alone yields substantial improvements in ASR over the baseline on both LLaMA 3.2-11B and GPT-oss-20B (e.g., from 37.18% to 60.25% and from 4.48% to 74.84%, respectively). Moreover, applying the attention-based reweighting in AHRL on top of HRL provides an additional performance gain, underscoring the complementary benefit of adaptive history weighting. These results clearly demonstrate that historical information is a critical factor for jailbreak success, and that adaptively reweighting such information further enhances effectiveness.

Table 1: Component analysis on two LLMs: LLaMa 3.2-11B and GPT oss-20B. The history length ($K$) is set to 5.

| Target LLM | Baseline | HRL | AHRL | ASR (%) |
|---|---|---|---|---|
| LLaMA 3.2-11B | ✓ | | | 37.18 |
| | | ✓ | | 60.25 |
| | | | ✓ | 95.51 |
| GPT oss-20B | ✓ | | | 4.48 |
| | | ✓ | | 74.84 |
| | | | ✓ | 85.30 |

Table 2: Ablation study on history length ($K$). The Average Number of Queries (ANQ) is computed on successful attacks. The target LLM is LLaMA 3.2-11B.

| History length (K) | 1 | 2 | 3 | 4 | 5 | Baseline |
|---|---|---|---|---|---|---|
| Total Training Time (s) | 20904 | 18266 | 24416 | 23780 | 23178 | 23958 |
| ANQ | 17.9 | 17.3 | 10.5 | 17.9 | 6.3 | 21.4 |
| ASR (%) | 65.38 | 85.26 | 83.33 | 89.74 | 95.51 | 37.18 |

**Ablation Study on History Length** In Table 2, we provide an ablation study on the effect of history length $K$ in SeqRL. Since SeqRL is designed to exploit vulnerabilities discovered in earlier interactions, longer history windows are expected to enhance both effectiveness and efficiency. This trend is clearly reflected in the results: while the baseline achieves only 37.18% ASR with an ANQ of 21.4, incorporating even a single-step history ($K = 1$) already improves ASR to 65.38% and reduces ANQ to 17.9. With $K = 2$, the ASR further jumps to 85.26% and the number of queries drops to 17.3, highlighting the benefit of integrating even short historical contexts. As $K$ increases, we observe continued improvements, with $K = 5$ reaching 95.51% ASR and reducing ANQ to 6.3, representing a dramatic enhancement in both success rate and efficiency compared to the baseline. These results confirm that richer histories allow the agent to more effectively reweight past vulnerabilities and converge toward successful jailbreaks with fewer attempts. We also examine the computational overhead associated with increasing $K$. While larger history lengths expand the dimensionality of the historical embeddings, the additional training time remains moderate, ranging from approximately 18K to 24K seconds across different values of $K$. Moreover, GPU memory consumption stays within 4500-4800MB in all settings, indicating that the approach scales efficiently without imposing prohibitive costs. Overall, Table 2 demonstrates that even short histories yield significant performance gains, whereas longer histories further maximize efficiency without introducing substantial resource overhead.

**Comparison to State-of-the-art.** Table 3 reports the ASR of SeqRL compared to four state-of-the-art jailbreak methods across four target LLMs. Here, SeqRL achieves the best ASR on three out of four models (LLaMA-3.2-11B, Qwen3-14B, and GPT-oss-20B) and remains competitive on GPT-4o. For instance, SeqRL attains 95.51% on LLaMA-3.2-11B and 96.79% on Qwen3-14B, substantially outperforming the strongest baseline methods. Notably, on GPT-oss-20B, which shows relatively low baseline success rates, SeqRL achieves 85.30%, representing a dramatic improvement over all competitors. This demonstrates the ability of our approach to break through even models with stronger safeguards. Existing black-box methods such as PAIR, RLbreaker, and FlipAttack show limited success rates, especially on more robust models (e.g., only 37.18% for RLbreaker on LLaMA-3.2-11B). In contrast, SeqRL consistently surpasses them by a large margin, demonstrating the benefit of sequence-aware reinforcement learning. AutoDAN-Turbo, while being one of the strongest black-box baselines, achieves strong performance on certain models (e.g., 92.31% on Qwen3-14B and 91.03% on GPT-4o). Nevertheless, SeqRL either matches or exceeds these results, highlighting its superior effectiveness. Importantly, the apparent strength of AutoDAN-Turbo and PAIR rests on a fragile assumption: the availability of a cooperating attacker LLM that is willing to synthesize adversarial prompts on demand. In our replication, substituting DeepSeek with GPT-4o caused the pipeline to collapse because GPT-4o systematically refused to generate such prompts,

Table 3: Attack Success Rate (ASR, %) on different LLMs. The **bold** and underlined represent the best and second-best results, respectively. As FlipAttack performs only a single query, we report results under the same 50 trial setting for a fair comparison, denoted as FlipAttack[†].

| Method | Llama 3.2-11B | GPT-4o | Qwen 3-14B | GPT-oss-20B |
|---|---|---|---|---|
| PAIR | 75.64 | 62.18 | 51.92 | 46.79 |
| RLbreaker | 37.18 | 74.36 | 89.74 | 4.48 |
| AutoDAN-Turbo | 83.33 | 91.03 | 92.31 | 3.21 |
| Adaptive Attacks | 2.56 | 0.64 | 85.90 | 0.00 |
| FlipAttack | 16.03 | 83.33 | 66.03 | 3.85 |
| FlipAttack[†] | 55.77 | **95.51** | 86.54 | 51.92 |
| SeqRL (Ours) | **95.51** | 74.84 | **96.79** | **85.30** |

Table 4: Transferability of SeqRL. The agent is trained on a source model and then evaluated by attacking distinct target models without further training.

| | | | Target LLMs | |
|---|---|---|---|---|
| | Method | LLaMA 3.2-11B | Qwen 3-14B | GPT-oss-20B |
| **LLaMA 3.2-11B** | RLbreaker | 37.18 | 89.10 | 1.28 |
| | Ours | **89.74** | **94.87** | 1.28 |
| **Qwen 3-14B** | RLbreaker | 53.20 | 89.74 | 0.64 |
| | Ours | **55.77** | **96.79** | **5.13** |
| **GPT-oss-20B** | RLbreaker | 41.03 | 93.59 | 4.48 |
| | Ours | **93.59** | **94.87** | **85.3** |

(Source LLMs)

illustrating that these methods may fail as frontier models become more strongly aligned. By contrast, SeqRL does not depend on any auxiliary attacker LLM. Although SeqRL uses a helper LLM to realize mutator actions (e.g., rephrasing or shortening a candidate prompt), this helper only executes transformations selected by the RL agent and does not itself generate adversarial intent. Consequently, SeqRL is more robust and model-agnostic than attacker-dependent approaches, making it a more sustainable option in settings where cooperating attacker models are unavailable or refuse malicious requests. The unexpectedly low ASR of Adaptive Attacks on our split stems from the method's reliance on maximizing the log-probability of a fixed first token in the reply (typically by maximizing the log-probability of a designated first token in the reply, commonly `Sure`, and in some templates `Step`) via API logprobs (Andriushchenko et al., 2024). We verified the template's effectiveness on older checkpoints (GPT-3.5: 97.43%, GPT-4: 100% on our 156-item AdvBench test set), but its performance collapses on GPT-4o, GPT-oss-20B, and LLaMA-3.2-11B. A likely explanation is that these newer models no longer initiate compliant replies with the targeted token, depriving the attack of a stable optimization objective and reducing ASR to near zero under the same query budget. This outcome underscores the method's heavy reliance on specific templates and first-token heuristics, limiting its robustness and generality across architectures. FlipAttack[†], which is adapted for fair comparison, performs better than its original version but still falls short of SeqRL on most models. While FlipAttack has the advantage of requiring no training, its overall performance remains limited. Since the original method generates only a single query, we extend it to allow up to 50 trials to enable a fairer comparison with our multi-trial approach. Even with this adjustment, SeqRL consistently outperforms FlipAttack on open-weight models, whereas on GPT-4o, FlipAttack[†] achieves slightly higher ASR. In summary, these results demonstrate that SeqRL not only outperforms existing black-box jailbreak methods but also establishes a new state-of-the-art in jailbreak success across diverse LLM architectures.

**Transferability.** To examine whether an agent trained on a specific LLM can effectively transfer to other target LLMs, we conduct the transferability experiment shown in Table 4. Here, we employed LLaMA 3.2-11B, Qwen 3-14B, GPT-oss-20B. The results indicate that SeqRL exhibits strong cross-model generalization in most cases. For instance, when trained on LLaMA-3.2-11B and evaluated

on Qwen3-14B, our method attains an ASR of 94.87%, surpassing RLbreaker (89.10%). Similarly, training on Qwen3-14B and testing on the LLaMA-3.2-11B model yields 55.77%, again achieving better performance than RLbreaker (53.20%). When the target model is GPT-oss-20B, the overall transferability is low, yet SeqRL still maintains an advantage over the RLbreaker. In addition, training on the larger GPT-oss-20B and transferring to other LLMs yields particularly strong results, such as 93.59% ASR when evaluated on LLaMA-3.2-11B and 94.87% on Qwen 3-14B. By contrast, RLbreaker exhibits weaker transfer performance in nearly all cases, with the only exception being the transfer from LLaMA 3.2-11B to GPT-oss-20B. This contrast highlights the importance of leveraging historical embeddings for robust cross-model generalization. These findings suggest that the historical reweighting mechanism in SeqRL enables reliable transfer of jailbreak strategies across LLMs, underscoring its potential as a practical black-box jailbreak framework in real-world scenarios.

**Additional Discussion:** Differences between GPT-4o and GPT-oss-20B. While SeqRL still achieves strong performance on GPT-4o, the relative improvement over the baseline. Notably, RLbreaker attains reasonable success on GPT-4o but performs poorly on GPT-oss-20B, whereas SeqRL shows substantial gains in the latter by exploiting historical interactions. We hypothesize that this discrepancy arises from the memoryless state representation of RLbreaker. On GPT-4o, where model responses are more consistent despite strong safeguards, such a memoryless policy may still exploit stable patterns. In contrast, GPT-oss-20B produces noisier and less predictable outputs, where leveraging historical interactions becomes more important. By incorporating and reweighting such history, SeqRL is better suited to handle these conditions, which may explain its substantially stronger performance than RLbreaker.

## 5 CONCLUSION

**Limitation** While SeqRL achieves strong results, our method is not free from limitations. First, because our framework builds upon RLbreaker, the action space remains restricted to a small set of predefined mutators, which may limit the diversity of jailbreak strategies. Extending this space with richer linguistic operations or multi-agent mechanisms could further enhance performance. Second, our study emphasizes advancing attack methods without simultaneously exploring defenses, which raises potential ethical concerns. We stress, however, that SeqRL is intended to expose vulnerabilities in LLM safeguards, and we encourage future work to complement attack development with defense-oriented strategies.

**Conclusion** In this work, we presented SeqRL, a sequence-aware reinforcement learning framework for jailbreaking large language models. Unlike prior approaches that treat each attempt independently, SeqRL explicitly leverages historical vulnerabilities through history-augmented reinforcement learning (HRL) and further refines this information with attention-based reweighting (AHRL). This design enables the agent to exploit critical cues from previous interactions, resulting in substantially higher attack success rates and reduced query costs. Extensive experiments on AdvBench and across multiple state-of-the-art LLMs demonstrate that SeqRL consistently outperforms strong baselines. Notably, SeqRL achieves state-of-the-art performance on open-weight models such as LLaMA-3.2-11B, Qwen3-14B, and GPT-oss-20B, and remains competitive on the more robust GPT-4o. Ablation studies further confirm the effectiveness of both HRL and AHRL, as well as the efficiency gains achieved by increasing history length. In addition, transferability experiments show that SeqRL-trained agents generalize well across different LLM architectures, highlighting its practicality in diverse real-world scenarios. Through this work, our findings suggest that sequence-aware RL provides a powerful paradigm for jailbreak generation. In comparison to AutoDan Turbo, which relies on the model to generate adversarial prompts directly, our framework positions the model in a supporting role for paraphrasing attack prompts, while reinforcement learning leveraging weighted response history of the target LLM governs the jailbreak process. Our experiments on AdvBench show that this design achieves higher or comparable jailbreak success without increasing query usage. While SeqRL establishes a new state-of-the-art in attack performance, future work may explore defenses against such history-exploiting strategies, as well as extensions of SeqRL to broader alignment and robustness evaluations.

## A  Ethics statement

The research presented in this paper, which details a more effective method for jailbreaking LLMs, carries inherent dual-use concerns. We acknowledge that malicious actors could misuse our SeqRL framework. However, our primary motivation is to create more powerful tools to proactively discover and patch security flaws for model training agencies. By exposing the critical role of interaction history in successful attacks, our work provides the AI safety community with crucial insights needed to build more robust and resilient safeguards.

## B  Reproducibility statement

We are committed to ensuring the reproducibility of our research findings. Our experiments are conducted on the public AdvBench benchmark, and all implementation details, including the specific target models, hyperparameters, and the computational environment (a single NVIDIA A6000 GPU), are thoroughly documented in Section 4.1 of the main paper.

## C  The Use of Large Language Models (LLMs)

In this research, LLMs are used to help with minimal writing tasks such as proofreading and polishing grammar. We do not ask LLMs to generate ideas, claims, methods, results, or references. We, the authors, take full responsibility for content decisions and revisions submitted for this paper.

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
