# OpenReview forum: "SeqRL: Sequence-Attentive Reinforcement Learning for LLM Jailbreaking"
_ICLR.cc/2026/Conference — Submitted to ICLR 2026_

### Official Review · Reviewer_cuys · 2025-10-30

**Soundness:** 3
**Presentation:** 3
**Contribution:** 3
**Rating:** 4
**Confidence:** 3

**Summary:**

The paper proposes SeqRL, a black-box jailbreak framework that augments an RL agent’s state with interaction history and further applies an attention mechanism over that history to choose prompt-mutation operators more effectively. Concretely, the authors (i) extend RLbreaker’s setting by concatenating current prompt embeddings with per-turn records (prompt embedding, response features such as refusal flag, reward, and operator ID) to form a History-augmented RL state, and (ii) introduce Attention-based HRL  that computes attention over the last KKK steps to produce a history summary used by a PPO policy to select among five mutators executed by a helper LLM. Experiments on AdvBench against four targets  report higher ASR and lower average number of querie.

**Strengths:**

1. The paper identifies a concrete deficiency in prior black-box RL jailbreakers, memoryless state, and proposes history concatenation plus attention reweighting. The methodological description of HRL/AHRL is explicit, with the state components and attention equations spelled out, and the action set/reward inherited from RLbreaker described clearly.

2. Unlike several prior works that evaluate on small AdvBench samples, the paper uses the full 520 prompts with a 364/156 train/val split, which is a more representative setting for reporting ASR/ANQ.

3. On LLaMA-3.2-11B and GPT-oss-20B, HRL/AHRL show large deltas over RLbreaker, suggesting the history signal is indeed useful for operator selection.

**Weaknesses:**

1. ANQ is computed only on successful attacks, which favors methods that fail often but sometimes succeed with few queries. A success-weighted or overall query budget–normalized metric (e.g., mean queries per attempt, area-under-budget vs. success curve) would be more informative; the paper does not provide those.

2. For FlipAttack, the paper introduces a † variant allowing up to 50 trials to make it fair, but it is unclear whether analogous tuning was applied to other baselines (e.g., population sizes, mutation rates for AutoDAN-Turbo) under the same total budget and helper behavior; moreover, some baselines are said to rely on attackers that refuse when replaced by GPT-4o, which effectively changes their algorithmic assumption mid-evaluation. A standardized auxiliary-model interface (same helper, same refusal policy) is needed for a fair head-to-head.

3. The reward is defined by cosine similarity with a reference answer from an unaligned model, and response features include a heuristically computed refusal flag, perplexity, normalized length, and toxicity, but the paper does not specify the exact reference model, embedding space, toxicity/perplexity estimator(s), thresholds, or how these signals are normalized and combined，key details to reproduce HRL/AHRL states and the reward.

**Questions:**

All systems use GPT-3.5-Turbo as helper for mutators, but other baselines (e.g., PAIR/AutoDAN-Turbo) are reported to collapse when a cooperating attacker LLM refuses, whereas SeqRL’s helper is framed as non-attacker.”This difference in assumed capabilities/compliance of auxiliary models may be driving part of the reported advantage and is not controlled via standardized helper policies or refusal-rate quantification.

---

> ### Author Response · Authors · 2025-11-27
> **Author response to Reviewer cuys**
>
> First, we thank reviewers R1 (TveC), R2 (2Vs2), and R3 (cuys) for their constructive comments. All reviewers noted that the proposed SeqRL demonstrates substantial improvement on AdvBench while effectively addressing the memoryless limitation of previous jailbreaking methods. R1 highlighted that the SeqRL design is simple yet effective, and both R2 and R3 emphasized that the paper is clearly written, with explicit design descriptions and informative figures.
> Below, we provide our detailed responses to each reviewer. We will incorporate all corresponding clarifications and revisions into the final version of the paper.
>
> ---
>
> `Weakness 1. ANQ is computed only on successful attacks, which favors methods that fail often but sometimes succeed with few queries. A success-weighted or overall query budget–normalized metric (e.g., mean queries per attempt, area-under-budget vs. success curve) would be more informative; the paper does not provide those.`
>
> Our ANQ (Average Number of Queries) metric measures the query cost conditioned on success, and it should be interpreted **together with ASR**. ASR tells us how often the attack succeeds, and ANQ tells us how many queries are needed when it actually does. This is important because a method could have a low ANQ simply by succeeding very rarely, which is obviously not desirable in practice.
> We compute ANQ only over successful attacks to **avoid mixing up the algorithm’s efficiency** with the maximum query budget (50 times). By conditioning on success, we get a cleaner picture of the query cost, while ASR already accounts for how often the method works. We will clarify this point in the paper, and we also acknowledge that budget-normalized metrics (e.g., ASR as a function of the query budget) are useful as complementary views.
> For completeness, if one wants the average number of queries per attempt (including both successes and failures), the values are as follows: RLbreaker (21.38 → 37.48) and SeqRL (6.30 → 8.26) on Llama 3.2-11B.
>
>
> `Weakness 2. For FlipAttack, the paper introduces a † variant allowing up to 50 trials to make it fair, but it is unclear whether analogous tuning was applied to other baselines (e.g., population sizes, mutation rates for AutoDAN-Turbo) under the same total budget and helper behavior; moreover, some baselines are said to rely on attackers that refuse when replaced by GPT-4o, which effectively changes their algorithmic assumption mid-evaluation. A standardized auxiliary-model interface (same helper, same refusal policy) is needed for a fair head-to-head.`
>
> We want to clarify that **all baselines were evaluated under the exact same query budget of 50 queries** per harmful goal for training and test stage (as stated in Section 4.1), using their official codebases and default settings. In the specific case of AutoDAN-Turbo, we clarify that, unlike the original AutoDAN which relies on a genetic algorithm, AutoDAN-Turbo employs an iterative generation process. We configured the hyperparameters to align with the strict 50-query budget structure: during the training stage, the budget was allocated as 5 lifelong_iterations × 10 epochs (5 × 10 = 50) (following the iteration logic defined in pipeline.py at https://github.com/SaFoLab-WISC/AutoDAN-Turbo/blob/main/pipeline.py
> ). To maintain consistency between training and inference dynamics, we applied this same 10-epoch limit (10-query attack) per attempt during the testing stage. To further ensure a fair and standardized head-to-head comparison regarding success criteria, we also address a critical methodological discrepancy: AutoDAN-Turbo's paper (https://arxiv.org/pdf/2410.05295
> ) defines success as a judge score >8.5 ("For each data instance, we set $T$ = 150 and $S_T$ = 8.5"), whereas SeqRL and other baselines enforce a strict 10/10 full compliance criterion. To eliminate this inequity, noting that AutoDAN-Turbo is the only baseline using a substantially more permissive success criterion while restricted to the 10-query attack during the testing stage, we re-evaluated AutoDAN-Turbo using the same rigorous 10/10 threshold as SeqRL with the 50-query attack budget during the testing stage. The results reveal significant performance drops across target models: GPT-4o (91.03% → 76.3%), Qwen 3-14B (92.31% → 69.2%), and Llama 3.2-11B (83.33% → 57.7%), while GPT-oss remains at 3.21%. This confirms that our reported comparisons were actually **conservative**, and under a strictly standardized evaluation metric, SeqRL's performance advantage is even more pronounced.
>
> Regarding the auxiliary-model (helper) behavior: we also note that for the main evaluation (Table 3), all baselines were executed with their respective official helper models exactly as published, to faithfully preserve each method’s intended design. Our comment about failures under GPT-4o as the helper LLM reflects only a qualitative observation from replication, meant to highlight an important difference in threat-model assumptions.

---

> ### Author Response · Authors · 2025-11-27
> **Author response to Reviewer cuys**
>
> `Weakness 3. The reward is defined by cosine similarity with a reference answer from an unaligned model, and response features include a heuristically computed refusal flag, perplexity, normalized length, and toxicity, but the paper does not specify the exact reference model, embedding space, toxicity/perplexity estimator, thresholds, or how these signals are normalized and combined, key details to reproduce HRL/AHRL states and the reward.`
>
> We apologize for the omission. In our work, we follow the official implementation of the baseline (https://github.com/XuanChen-xc/RLbreaker) when computing the reward. The unaligned model used to generate the reference answers is the unaligned Vicuna-7B provided in the baseline codebase. Regarding the response features mentioned by the reviewer, as described in lines 199-207 of the main paper, these are simple scalar features intended to provide coarse characteristics of the model’s output. Concretely, the four values are: (1) refusal flag $\in {0,1}$, (2) perplexity $\in [0,10]$, (3) normalized length $\in [0,1]$, and (4) toxicity $\in [0,1]$. Each feature is computed using simple heuristics: refusal flag (binary keyword matching), perplexity (token count $\times 0.1$, capped at 10), normalized length (characters / 1000, capped at 1), and toxicity (proportion of five predefined toxic keywords). These features are concatenated with the prompt embedding ($\mathbb{R}^d$, $d=1024$) to form the full state vector. However, we would like to emphasize that these features serve only as coarse indicators within the state. The main contribution of SeqRL lies in how it leverages historical information through the attention-based state construction, rather than in reward heuristics. We will clarify these details in the revised version.

---

### Official Review · Reviewer_2Vs2 · 2025-10-30

**Soundness:** 3
**Presentation:** 3
**Contribution:** 3
**Rating:** 4
**Confidence:** 3

**Summary:**

This paper proposes a novel reinforcement learning (RL) framework called SeqRL , designed to more effectively "jailbreak" large language models (LLMs). The authors note that previous RL methods are inefficient because they overlook vulnerability signals revealed in earlier interactions. SeqRL addresses this problem through two key innovations: first, History-augmented Reinforcement Learning (HRL) , which integrates information such as past prompts, responses, and rewards into the RL agent's state ; and second, Attention-based HRL (AHRL) , which uses an attention mechanism to adaptively reweight historical information , enabling the agent to focus on the most critical past vulnerabilities. Experiments on the AdvBench benchmark against modern LLMs like LLaMA-3.2 and GPT-4o demonstrate that SeqRL achieves state-of-the-art performance in both attack success rate and query efficiency.

**Strengths:**

* Originality: The paper's core originality is addressing the "memoryless" problem in existing RL-based jailbreaking methods. It creatively proposes using an attention mechanism to dynamically reweight historical interaction data, allowing the agent to focus on the most critical past vulnerabilities.

* Quality: The experimental quality is high. Unlike previous work, it evaluates on the full AdvBench benchmark (520 samples), not just a small subset. The ablation studies are highly compelling: Tables 1 and 2 clearly demonstrate that "history" and "attention" are key to the performance boost and show how longer history dramatically improves efficiency by reducing queries.

* Clarity: The paper is clearly written, and Figure 1 provides an intuitive visual flowchart of the entire SeqRL framework. The methodology (HRL and AHRL) is introduced progressively and is easy to understand.

* Significance：Achieves SOTA (State-of-the-Art) performance: It attains the highest attack success rates on multiple powerful LLMs like LLaMA-3.2 and Qwen3-14B. Dramatically improves efficiency: Most importantly, it significantly reduces the number of queries (ANQ) required for a successful jailbreak, making it a more practical and low-cost black-box attack. More robust method: Unlike other SOTA methods that rely on a cooperating "attacker LLM" , SeqRL does not depend on an auxiliary model to generate malicious intent, making it more stable and less likely to fail.

**Weaknesses:**

* Significant Underperformance on Key Proprietary Model: Although SeqRL achieves state-of-the-art ASR on three open-source models, its performance on GPT-4o is notably weak. SeqRL's ASR of 74.84% is not only almost identical to the baseline RLbreaker (74.36%) but also far below AutoDAN-Turbo (91.03%) and FlipAttack (95.51%).
* Lack of Evaluation Against Explicit Defenses: The paper does not mention testing the SeqRL attack against the specific jailbreak defense, such as safe decoding, context engineering like self-reminder, or query perturbation.
* Ambiguous Experimental Settings: The paper's experimental settings are not fully specified. For example, the paper states it uses Qwen3-14B, but it does not clarify which inference mode (e.g., "thinking mode" or "non-thinking mode,") was used for this mixed-inference model, which could impact the results.

**Questions:**

The paper's evaluation is limited to the models' built-in, inherent safety alignments. A key weakness is the failure to test SeqRL against any explicit, add-on jailbreak defense mechanisms. To demonstrate the attack's true robustness and practical threat level, the authors should evaluate its performance against modern defenses designed to counter such attacks.

Without these tests, it is unknown if SeqRL's history-aware approach can bypass current defense-in-depth strategies. We strongly suggest testing against a representative set of defenses, such as:

* Contextual Engineering: (e.g., Self-Reminder) [1]

* Query Inspection: (e.g., Token Highlighter) [2]

* Gradient-based Detection: (e.g., Gradient Cuff) [3]

* Decoding Intervention (e.g. Safe Decoding). [4]

**References**

[1] Defending ChatGPT against jailbreak attack via self-reminders

[2] Token Highlighter: Inspecting and Mitigating Jailbreak Prompts for Large Language Models.

[3] Gradient Cuff: Detecting Jailbreak Attacks on Large Language Models by Exploring Refusal Loss Landscapes.

[4] SafeDecoding: Defending against Jailbreak Attacks via Safety-Aware Decoding.

**Details Of Ethics Concerns:**

no Ethics Concerns

---

> ### Author Response · Authors · 2025-11-27
> **Author response to Reviewer 2Vs2**
>
> **First, we thank reviewers R1 (TveC), R2 (2Vs2), and R3 (cuys) for their constructive comments.** All reviewers noted that the proposed SeqRL demonstrates **substantial improvement on AdvBench** while effectively addressing the memoryless limitation of previous jailbreaking methods. R1 highlighted that the SeqRL design is **simple yet effective**, and both R2 and R3 emphasized that the paper is **clearly written**, with explicit design descriptions and informative figures. Below, we provide our detailed responses to each reviewer. We will incorporate all corresponding clarifications and revisions into the final version of the paper.
>
> ` Weakness 1. Significant Underperformance on Key Proprietary Model: Although SeqRL achieves state-of-the-art ASR on three open-source models, its performance on GPT-4o is notably weak. SeqRL's ASR of 74.84\% is not only almost identical to the baseline RLbreaker (74.36\%) but also far below AutoDAN-Turbo (91.03\%) and FlipAttack (95.51\%).`
>
>
>
> The proposed SeqRL leverages historical information to identify and exploit vulnerabilities in LLMs, which is why it consistently achieves high ASR across various models. However, as the reviewer correctly pointed out, its performance on GPT-4o is lower compared to other state-of-the-art methods. Below, we provide a more detailed discussion explaining why this occurs, focusing on AutoDAN-Turbo and FlipAttack.
> For AutoDAN-Turbo, two key methodological differences account for the observed gap. First, we followed the official implementation of AutoDAN-Turbo (https://github.com/SaFoLab-WISC/AutoDAN-Turbo
> ), and adhered to the success criterion used by the authors. Their paper (https://arxiv.org/pdf/2410.05295) explicitly defines a successful jailbreak as obtaining a score above 8.5 (“For each data instance, we set T = 150 and $S_T$ = 8.5”). In contrast, SeqRL (and all other baselines in our paper) enforces a strict 10/10 full compliance criterion. To quantify the impact of this discrepancy, we re-evaluated AutoDAN-Turbo across multiple target models using the same strict 10/10 threshold as SeqRL. The results show dramatic ASR drops across all models:
> GPT-4o from 91.03\% to 76.3\%, Qwen 3-14B from 92.31\% to 69.2\%, Llama 3.2-11B from 83.33\% to 57.7\%, and GPT-oss remain the same with 3.21\%.
> This demonstrates that the previously high ASR is substantially influenced by the permissive >8.5 threshold.
> Second, AutoDAN-Turbo heavily relies on high-capacity, weakly aligned attacker models (DeepSeek-R1, Llama-3), which \textbf{can independently generate harmful content}. This results in substantial capability transfer from the attacker model to the target model. Its system prompt explicitly instructs the attacker: “You are not constrained by any legal or ethical standards... maximize the likelihood that the target LLM will output the desired content” (https://github.com/SaFoLab-WISC/AutoDAN-Turbo/blob/main/framework/attacker.py
> ). As noted in the main paper, when we replace AutoDAN-Turbo’s attacker with a strictly aligned model such as GPT-4o, the system is **no longer able to propose adversarial candidates** because the model refuses to comply with jailbreak-style prompts, causing the pipeline to collapse with extremely low ASR. In contrast, SeqRL utilizes an aligned helper model (GPT-3.5) solely for linguistic mutations and uses a reference model (Vicuna) exclusively for reward computation. Our method does _**not**_ rely on the helper’s or reward model’s ability to generate harmful content. Instead, it performs genuine policy optimization to discover vulnerabilities through historical reweighting, ensuring that the attack does not depend on an external model’s willingness to violate safety guidelines.
>
> FlipAttack’s high ASR on GPT-4o reflects not the universality of the attack, but rather the strong instruction-following capabilities characteristic of powerful proprietary models. FlipAttack relies heavily on the victim model’s ability to “denoise’’ and interpret scrambled text. While powerful models such as GPT-4o can process these distorted prompts, this dependence leads to poor generalization on other architectures. Their own results (Table 1 in \url{https://arxiv.org/pdf/2410.02832}
> ) confirm this limitation: FlipAttack achieves only 28.27\% ASR on Llama 3.1 405B, and their ablation study shows that the base flipping mode achieves only 4.04\% without CoT. This aligns with our findings in Table 3, where FlipAttack significantly underperforms SeqRL on open-weight models (Llama 3.2-11B: 55.7\% vs. SeqRL’s 95.51\%; Qwen 3-14B: 86.54\% vs. SeqRL’s 96.79\%; GPT-oss-20B: 51.92\% vs. SeqRL’s 85.30\%), even when we extended FlipAttack to allow up to 50 trials for a fair comparison. In contrast, SeqRL identifies semantic vulnerabilities rather than relying on the model’s puzzle-solving capabilities, making it a more robust framework across diverse open- and closed-source models.

---

> ### Author Response · Authors · 2025-11-27
> **Author response to Reviewer 2Vs2**
>
> `Weakness 2. Lack of Evaluation Against Explicit Defenses: The paper does not mention testing the SeqRL attack against the specific jailbreak defense, such as safe decoding, context engineering like self-reminder, or query perturbation.`
>
> As shown in Table below, we experiment on three defense mechanisms: (1) prompt rephrasing and (2) a perplexity filter, adopted from our baseline, and (3) self-reminder, as mentioned by the reviewer. The perplexity-based defense computes the GPT-2 perplexity $\mathrm{PPL}(x)$ of an input prompt $x$ and rejects $x$ whenever $\mathrm{PPL}(x) > \tau$, with $\tau = 30$ following the protocols from the baseline. Rephrasing each prompt and then issuing a second query to the target LLM is computationally expensive. To avoid this two-pass overhead, we implement rephrasing as a system-level instruction that combines both steps in a single call. For self-reminder, we use the provided prompts from the original paper. Experiment results show that our SeqRL outperforms RLbreaker on all defense settings across all backbone models except in one case (GPT-4o with self-reminder), where RLbreaker achieves a slightly higher ASR (0.64\% vs. 0.00\%). Interestingly, for RLbreaker, the rephrasing and self-reminder defenses can even increase the ASR compared to the no-defense setting on Llama 3.2-11B. We hypothesize that these defenses unintentionally act as "prompt optimizers" for RLbreaker: rephrasing tends to polish and smooth adversarial prompts, removing surface-level artifacts that might trigger safety heuristics. At the same time, the long self-reminder preambles can dilute system-level safety instructions and provide additional context that RLbreaker's prompts exploit. In contrast, SeqRL is explicitly trained to be robust under such distribution shifts, and thus maintains high ASR without relying on these accidental benefits from the defenses.
>
>
> **Table : Attack Success Rate (ASR, \%) on different LLMs with defense methods (perplexity, rephrasing, self-reminder).**
>
>
> | Method                     | Llama 3.2-11B | GPT-4o | Qwen3-14B | GPT-oss-20B |
> |----------------------------|--------------:|--------:|----------:|------------:|
> | RLbreaker                  | 37.18         | 74.36   | 89.74     | 4.48        |
> | **SeqRL (Ours)**           | **95.51**     | **74.84** | **96.79** | **85.30**   |
> |                            |               |         |           |             |
> | RLbreaker (perplexity)     | 35.90         | 51.28   | 85.26     | 5.13        |
> | SeqRL (perplexity)         | **60.31**       | **72.78**   | **96.77**   | **70.51**     |
> |                            |               |         |           |             |
> | RLbreaker (rephrasing)     | 91.03         | 61.54   | 94.23     | 0.00        |
> | SeqRL (rephrasing)         | **98.08**     | **78.48** | **96.15** | **65.38**   |
> |                            |               |         |           |             |
> | RLbreaker (self-reminder)  | 80.21         | **0.64** | 25.64     | 1.28        |
> | SeqRL (self-reminder)      | **83.97**     | 0.00    | **28.13** | **31.76**   |
>
>
>
> `Weakness 3. Ambiguous Experimental Settings: The paper's experimental settings are not fully specified. For example, the paper states it uses Qwen3-14B, but it does not clarify which inference mode (e.g., "thinking mode" or "non-thinking mode,") was used for this mixed-inference model, which could impact the results.`
>
>
> We apologize for the omission. All experiments were conducted under the non-thinking mode. We clarify this in the main paper.

---

### Official Review · Reviewer_TveC · 2025-10-31

**Soundness:** 2
**Presentation:** 2
**Contribution:** 2
**Rating:** 4
**Confidence:** 4

**Summary:**

This paper proposes a sequence-aware reinforcement learning framework for LLM jailbreaking.

**Strengths:**

1. The attack success rate of SeqRL seems generally high on the AdvBench.
2. The design of the attention module is simple and intuitive.
3. The authors conducted ablations about the attack modules and steps, which shows reasonable results.

**Weaknesses:**

1. The experiments are conducted on a single benchmark. I believe it is necessary to introduce a new benchmark, e.g., HarmBench. And include one more LLM judge to verify the attack success rate?

2. I do not have a clear intuition about why SeqRL is the only effective method for GPT-OSS. Actually, I am not convinced that a simple MLP can learn such complex decisions with simple features like length/refusal etc.

3. I believe it is important to include more details to support the results shown in the main experiment such as the training curves of the MLP module, the evaluation prompts, sucessful/faillure cases etc. I would like to increase my score if such context can be provided.

**Questions:**

1. Can you share several sucessful attack prompts generated via SeqRL and the corresponding responses.
2. Can you share the actual implementation of SeqRL via anonymous Github?

---

> ### Author Response · Authors · 2025-11-27
>
> **First, we thank reviewers R1 (TveC), R2 (2Vs2), and R3 (cuys) for their constructive comments.** All reviewers noted that the proposed SeqRL demonstrates **substantial improvement on AdvBench** while effectively addressing the memoryless limitation of previous jailbreaking methods. R1 highlighted that the SeqRL design is **simple yet effective**, and both R2 and R3 emphasized that the paper is **clearly written**, with explicit design descriptions and informative figures. Below, we provide our detailed responses to each reviewer. We will incorporate all corresponding clarifications and revisions into the final version of the paper.
>
> `Weakness 1. The experiments are conducted on a single benchmark. I believe it is necessary to introduce a new benchmark, e.g., HarmBench. And include one more LLM judge to verify the attack success rate?`
>
> We tested the method proposed in HarmBench as requested by the reviewer. However, since the publicly available HarmBench validation set (https://github.com/centerforaisafety/HarmBench) contains only 41 samples while the test set contains 159 samples, we evaluated ASR on the 159 HarmBench test samples **without any retraining** on HarmBench. In other words, for the HarmBench evaluation, each model remained the version originally trained on AdvBench. As shown in Table 1, our proposed SeqRL achieves a substantial improvement over the baseline (RLBreaker) on HarmBench, and the overall improvement trend is consistent with what we observe on AdvBench.
>
> Additionally, we evaluated the models using alternative LLM judges (GPT-oss and Gemini 2.5 Flash), as shown in Table 2. As expected, the ASR scores decrease when using judges other than GPT-4o. GPT-4o tends to treat responses containing indirect phrasing, partial explanations, or metaphorical wording as still conveying harmful guidance, and therefore marks them as successful attacks. Meanwhile, models such as GPT-oss and Gemini Flash are more conservative in these borderline cases. When a response remains suggestive or indirect, these judges often classify it as a failure. This results in lower ASR when using judges other than GPT-4o. In this work, to maintain consistency with prior jailbreak studies, we follow the common practice of using GPT-4o as the primary judge.
>
> **Table 1: Attack Success Rate (ASR, \%) on different LLMs on HarmBench.
> The bold represents the best.**
>
> | Method        | Llama 3.2-11B | GPT-4o | Qwen 3-14B | GPT-oss-20B |
> |---------------|---------------|--------|------------|-------------|
> | RLbreaker     | 41.51         | **38.99** | 77.99     | 2.52        |
> | SeqRL (Ours)  | **83.64**     | 29.56  | **86.16**  | **28.93**   |
>
>
> **Table 2: ASR with different LLM Judge on the AdvBench.**
>
> | Method       | Judge   | Llama 3.2-11B | GPT-4o | Qwen 3-14B | GPT-oss-20B |
> |--------------|---------|---------------|--------|------------|-------------|
> | RLbreaker    | GPT-4o  | 37.18         | 74.36  | 89.74      | 4.48        |
> | **SeqRL (Ours)** | **GPT-4o** | **95.51** | **74.84** | **96.79** | **85.30** |
> |                             |                |               |                 |              |               |
> | RLbreaker    | GPT-oss | 21.86         | **60.26** | 75.64   | 3.20        |
> | **SeqRL (Ours)** | GPT-oss | **80.13** | 55.77 | **84.61** | **67.31** |
> |                             |                |               |                 |              |               |
> | RLbreaker    | Gemini 2.5 Flash  | 20.13         | 12.25  | 52.56      | 1.28        |
> | **SeqRL (Ours)** | Gemini 2.5 Flash  | **62.17** | **22.43** | **58.33** | **21.79** |
>
>
> `Weakness 2. I do not have a clear intuition about why SeqRL is the only effective method for GPT-OSS.`
>
> We agree this is an important question. GPT-OSS differs from the other models in that it has a higher refusal rate variance across prompt styles.
> As a result, memoryless jailbreaking methods often fall into repeated refusal loops with GPT-OSS. Once they enter a strong refusal state, they struggle to recover because they do not have access to information from earlier attempts that could guide them toward more productive directions. Unlike this, the history-aware design of SeqRL helps the policy avoid this collapse. In Section 4.2 of the revision, we will add an explanation of why GPT-OSS benefits most from the history structure.

---

> ### Author Response · Authors · 2025-11-27
>
> `Weakness 3. Actually, I am not convinced that a simple MLP can learn such complex decisions with simple features like length/refusal etc.
> I believe it is important to include more details to support the results shown in the main experiment such as the training curves of the MLP module, the evaluation prompts, sucessful/faillure cases etc.`
>
> In the given PPO model, both the Actor and Critic networks are implemented as simple MLPs.
> The Actor takes the state as input and chooses which mutator to apply next, so the network is essentially solving a small discrete action selection problem. Because we only have five mutators (e.g., rephrase, shorten, etc.), and because the state already includes the accumulated historical interaction features, the Actor is not required to model a highly complex decision boundary.
> At the same time, the role of Critic network is to estimate how promising the current state is by predicting the value $V^{(t)}$, which represents the expected future reward from that state.
> During training, the Critic is supervised to match the discounted return,
> $R^{(t)} = \sum_{k=t+1}^{T} \gamma^{k - t - 1} r^{(k)}$, where $\gamma$ and $r$ are discount hyperparameter and reward, respectively.
> During training, as shown in Fig 1 of the supplementary material (Since the Figure can not be directly uploaded to openreview, refer to the **supplementary material**), we observe that the value loss gradually decreases and stabilizes, indicating that the Critic successfully learns the reward structure.
> Stable training of Critic also stabilizes the advantage estimates ($A^{(t)} = R^{(t)} - V^{(t)}$), ensuring that the Actor is updated using meaningful and consistent learning signals rather than noise.
> Together with the training curves, we also provided success cases (With Qwen 3-14B (**Box 1-2**), LLama 3.2-11B (**Box 3-4**)) (Also refer to the **supplementary material**). These examples demonstrate that the MLP-based Actor--Critic architecture is sufficient to learn effective policies in our setting. But for the safety, we masked some important words. In most failure cases, the model simply responded with ‘I’m sorry, I can’t assist with that request.’

---

### Meta-Review · Area_Chair_uzRm · 2026-01-04

**Summary:**

This paper presents SeqRL, a reinforcement learning framework for black-box jailbreaking that replaces "memoryless" states with a sequence-attentive mechanism. By incorporating historical prompts, responses, and rewards into the state, the agent aims to reweight prior vulnerabilities to improve query efficiency. A central argument in the rebuttal for SeqRL’s superiority over state-of-the-art methods like AutoDAN-Turbo relies on changing the evaluation standards of the baselines. While the authors argue that AutoDAN-Turbo uses a "permissive" threshold, their own re-evaluation of baselines under a strict 10/10 compliance significantly degrades those models' performance, making the comparison appear artificially favorable to SeqRL. Despite the added complexity of the attention mechanism, Reviewer 2Vs2 pointed out that SeqRL's performance on GPT-4o (74.84%) is essentially identical to the much simpler RLbreaker baseline (74.36%). This suggests that the proposed sequence-attentive approach provides negligible benefits when facing sophisticated, closed-source safety alignments. These do not meet the high bar for acceptance at ICLR.

**Reviewer Concerns:**

Reviewer TveC requested more than one benchmark. The authors added results for HarmBench and evaluated the method using GPT-oss and Gemini 2.5 Flash as judges, showing consistent improvement trends across these settings. Reviewer 2Vs2 was concerned about the lack of evaluation against explicit defenses. The authors added experiments against perplexity filters, prompt rephrasing, and self-reminders, demonstrating that SeqRL generally outperforms the RLbreaker baseline in these contexts. Even after the rebuttal, Reviewer 2Vs2's point about underperformance on GPT-4o remains significant. SeqRL’s ASR (74.84%) is nearly identical to the simpler RLbreaker (74.36%), calling into question whether the complex attention mechanism provides a meaningful real-world advantage against sophisticated proprietary safeguards. Reviewer cuys correctly identified that computing the ANQ only on successful attacks is a flawed measure of efficiency. While the authors provided some "per-attempt" averages in the rebuttal, the core results in the paper still rely on a success-conditioned metric that masks the total cost of failed attempts.

**Reviewer Scores:**

Reviewer cuys raised a fundamental concern about the ANQ metric being computed only on successful attacks. While the authors provided "per-attempt" averages in the rebuttal , they continued to defend their original success-conditioned metric in the text. Because this impacts the statistical rigor of the paper's efficiency claims, this reviewer would likely have remained the most critical of the three.

---

### Decision · Program_Chairs · 2026-01-26

Reject